# Current Main Topics in Multiple Myeloma

**DOI:** 10.3390/cancers15082203

**Published:** 2023-04-08

**Authors:** Sonia Morè, Laura Corvatta, Valentina Maria Manieri, Attilio Olivieri, Massimo Offidani

**Affiliations:** 1Clinica di Ematologia Azienda Ospedaliero, Universitaria delle Marche, 60126 Ancona, Italy; 2Unità Operativa Complessa di Medicina, Ospedale Profili, 60044 Fabriano, Italy

**Keywords:** multiple myeloma, minimal residual disease, monoclonal antibodies, CAR T cell, bispecific antibodies

## Abstract

**Simple Summary:**

In newly diagnosed multiple myeloma patients (NDMM) the introduction of three-drug, and recently, four-drug combinations allowed to reach response rates never seen before, leading to significantly improved PFS and OS. Long-term therapies play a key role in delaying or preventing relapses, but they are expensive and may cause significant toxicities. As a result, several ongoing trials are evaluating the possibility to intensify or de-intensify treatment based on minimal residual disease status, assessed by highly sensitive molecular or immunophenotypic methods. In relapsed/refractory patients (RRMM), especially those with advanced disease who become refractory to all available agents, new generation immunotherapies, such as conjugated monoclonal antibodies (mAbs), bispecific antibodies and CAR-T cells showed relevant results. In patients with high-risk cytogenetics, outcome remains poor and results from risk-adapted strategies are not yet available. Here we discuss the most recent issues regarding the management of MM, reporting the most up-to-date modalities of treatment and monitoring under evaluation.

**Abstract:**

Multiple Myeloma (MM) remains a difficult to treat disease mainly due to its biological heterogeneity, of which we are more and more knowledgeable thanks to the development of increasingly sensitive molecular methods that allow us to build better prognostication models. The biological diversity translates into a wide range of clinical outcomes from long-lasting remission in some patients to very early relapse in others. In NDMM transplant eligible (TE) patients, the incorporation of mAb as daratumumab in the induction regimens, followed by autologous stem cell transplantation (ASCT) and consolidation/maintenance therapy, has led to a significant improvement of PFS and OS.; however, this outcome remains poor in ultra-high risk MM or in those who did not achieve a minimal residual disease (MRD) negativity. Several trials are exploring cytogenetic risk-adapted and MRD-driven therapies in these patients. Similarly, quadruplets-containing daratumumab, particularly when administered as continuous therapies, have improved outcome of patients not eligible for autologous transplant (NTE). Patients who become refractory to conventional therapies have noticeably poor outcomes, making their treatment a difficult challenge in need of novel strategies. In this review, we will focus on the main points regarding risk stratification, treatment and monitoring of MM, highlighting the most recent evidence that could modify the management of this still incurable disease.

## 1. Introduction

Although MM currently remains an incurable disease, therapeutic strategies developed over time have led to impressive improvement in overall survival (OS), suggesting that some patients could be considered cured [1]. However, both inter-patient and intra-patient heterogeneity constitute the biggest obstacle in treating and curing MM, even though innovative approaches to either newly diagnosed or in relapsed/refractory patients have tried to overcome them. In upfront settings, continuous or long-term therapies including mAbs as daratumumab have recently shown to induce deep and sustained responses [2,3,4]; however, it remains unclear what the optimal duration of treatment may be or how to minimize toxicities. In this regard, MRD status has shown to be not only a predictor of progression free survival (PFS) and OS but also an extremely valid biomarker to design MRD-driven therapies and to guide the duration of treatment. Although ASCT still plays a significant role in patients who are eligible, in the near future the unprecedented responses seen with quadruplet combinations including mAbs could modify this paradigm. However, moving novel effective drugs as proteasome inhibitors (PIs), immunomodulatory agents (IMiDs) and mAbs in the upfront setting means that the number of patients becoming early refractory to them is growing [5]. Luckily, newly developed immunotherapies like conjugated antibodies, bispecific antibodies or CAR T cell therapies are representing a turning point in the management of heavily pre-treated and refractory MM patients, despite many issues regarding their use that still need to be solved. In this review, we summarized the more relevant topics regarding the approach to patients with MM with a focus on the most recent clinical data.

## 2. Improvements in Risk Stratification

Despite the application of validated risk stratification systems, the category of intermediate risk MM now includes a very heterogeneous group of patients [6], characterized by vastly different prognoses, opening the challenge to build a more exhaustive risk stratification model [7].

The three-stage classification International Staging System (ISS) [8], which combines serum β2-microglobulinemia and albumin, depicting disease burden, has proven to be the simplest, most powerful and reproducible staging system and demonstrated to be more effective in comparison to the Durie and Salmon one [9]. Revised-ISS (R-ISS) incorporated two further prognostic factors: the presence of high risk genetic mutations [del(17p), t(4;14), or t(14;16)], assessed by fluorescence in situ hybridization (FISH), and lactate dehydrogenase level (LDH), as representative of disease burden [10]. It is currently used to prognosticate NDMM patients, it can predict early post-transplant relapse and have an independent prognostic effect on post-relapse survival after an early relapse [11]. The main limitation of the R-ISS was that 62% of patients were classified as intermediate-risk (R-ISS II), possibly including patients with different risk levels of progression/death. Recently, D’Agostino et al. validated an improvement of R-ISS in R2-ISS [12], whereas Abdallah published the Mayo Additive Staging System (MASS) [13], by adding 1q gain/amplification among the high risk cytogenetic features and considering the prognostic meaning of each single baseline risk feature in an additive fashion. D’Agostino et al. [12] validated R2-ISS in 10,843 NDMM patients, enrolled in clinical trials from 2005 to 2016, in the context of the HARMONY Project, and identified four groups of patients: R2-ISS I (19.2%), II (30.8%), III (41.2%) and IV (8.8%), redeploying intermediate-risk population in two different and more precise stages. Outcomes were significantly different among these groups, above all between the two intermediate groups, both in TE (OS 140 months in R2-ISS II and 75 months in R2-ISS III) and in NTE (OS 66 months in R2-ISS II and 52 months in R2-ISS III) patients, and independently of the upfront therapy they have received. This new score considered ISS and LDH, as previously, but it took a step forward regarding the karyotype by considering del(17p), t(4;14) and the role of the combination of different adverse cytogenetic features [14]. It did not account t(14;16) that was listed among the unfavourable high-risk chromosomal abnormalities by IMWG [15,16,17], because it was demonstrated to be significant in terms of OS, but not PFS, rare and usually present together with other adverse prognostic features [18,19,20]. Differently from previous scores, abnormalities of 1q were included in the R2-ISS, as also recommended by NCCN MM Panel [21], without a distinction between gain (3 copies) and amplification (>3 copies), although this would further improve the risk stratification. It was recently described that amp1q and its main clone position could be significantly associated with prognosis results, in comparison with gain1q [22,23,24,25], even if the biological meaning of amp1q remains difficult to determine. A recent retrospective experience confirmed the greater role of the association of high risk feature rather than a single one, but also the size of high risk clone influenced prognosis [26]. M-Smart MM risk stratification guideline by Mayo Clinic proposed an innovative concept, with patients possessing two of the high risk genetic abnormalities defined as “double hit” and having any three as “triple hit” MM [27]. Both phenotypes were considered ultra-high-risk disease characteristics, having observed that they correlated with aggressive clinical presentations, high early mortality and poor outcomes even in real life settings [9,28,29,30]. With these innovations—whose applications still have some difficulties when applied to the clinical practice setting because of the heterogeneity among labs and the lack of techniques’ standardization—a guideline update is needed.

Additionally, great advances in elucidating high-risk MM features have been achieved in recent years in terms of genomics (CoMMpass study) [31], but these advances are often difficult to translate into clinical practice. Whole-exome and whole-genome sequencing, made possible by the application of next generation sequencing techniques, have recently described new high-risk signatures [32], highlighting that del(17p)/TP53 mutations as well as 1q amplification—burden of driver gene and overall somatic missense mutation—are powerful drivers of poor prognosis, loss of heterozygosity (LOH) and an APOEBEC signature impact prognosis, as well as genomic clusters and genomic pairings, leading to the existence of “double hit” genotypes and dictate prognosis [30,33,34]. Moreover, many novel driver and oncogenic genes remain to be explored. Some gene expression profile (GEP) scoring systems were developed to better design a risk stratification, but only two have been validated: EMC92/SYK92 and UAMS GEP70 [35,36,37]. Genomic novel risk scores are detailed in Table 1.

In order to make prognostic parameters less difficult to technically calculate, Chanukuppa et al. recently found, through proteomic approaches, several dysregulated proteins, such as haptoglobin, kininogen 1, transferrin, and apolipoprotein A1, that could show a practical potentiality as early diagnostic and prognostic markers, after a validation in larger cohorts of MM patients [49]. Recent studies provided strong evidence that mass spectrometry (MS) should become a new standard to monitor monoclonal protein and, in combination with baseline disease features, improve prognostication in MM [50,51,52].

However, novel metabolomic approaches are still behind in the development process and need to be further investigated.

Considering the prognostic role of clinical features and disease burden, it is widely known that extramedullary localization of MM (EMD) at diagnosis confers a poor prognosis [53,54,55], but it has not been translated in a scoring system [56]. Garcès et al. recently demonstrated that quantification of circulating tumor cells (CTCs) in peripheral blood of transplant-eligible patients is a relevant prognostic factor at diagnosis, as a surrogate of both disease burden and trafficking, and provided cut-offs of >0.01% for the application of this biomarker in clinical practice [57]. They demonstrated that CTCs, detected by multi-flow cytometry, outperformed the quantification of plasma cells in bone marrow, that had historically a modest prognostic value [58,59,60,61]. Kostopoulos et al. [62] demonstrated in a real-life trial the negative and independent prognostic impact of increased CTC levels in the NDMM. This could be an easier prognostic tool to use and less invasive for patients to calculate, with the dilemma of the cut-off value of CTCs that is different among studies, ranged from 10 CPC per 50,000 events (0.02%) to 400 CPC per 150,000 events (0.267%), making difficult the clinical applicability. Some prognostic models are detailed in Table 1.

Rasche et al. [63] demonstrated the presence of ≥3 large (≥2.5 cm) focal lesions, described as areas of plasma cells by diffusion-weighted MRI with background suppression, was associated with poor outcome in NDMM, independently of R-ISS, GEP70, and extramedullary disease. It was also demonstrated that the genomic characterization into the focal lesion was very different from the iliac-crest derived plasma cells, harboring high risk subclones locally [64]. This special heterogeneity could explain why current prognostication models fail to describe the real prognosis of each MM patient, paving the way for a scoring system that incorporates different information coming from different disease sites. FDG-PET was used to identify, by the IMPeTUs criteria, focal lesions at baseline or the presence of hypermetabolic soft tissue components affecting survival in NDMM patients underwent to autologous stem cell transplant, irrespective of ISS stage or high risk cytogenetics [65]. Hovever, it needs further validations in order to use it in newer prognostic systems.

Lastly, Intzes et al. recently demonstrated socio-economic status (SES) as a surrogate prognostic marker for OS in MM patients, reflecting social aspects such as ethnicity/race, insurance coverage, place of living and accessibility to health services [66], complicating the challenging tangle of MM prognostic evaluation.

## 3. Current and Future Role of Immunotherapy in the Upfront Setting

Triplet-based regimens as induction therapy in TE patients have been found to be more effective than doublets [67], but, in the 2021 ESMO guidelines [68], only VRD (bortezomib, lenalidomide, dexamethasone) regimen is still recommended as first option in these patients. The phase III PETHEMA/GEM2012 trial evaluating VRD for 6 cycles followed by ASCT conditioned with busulfan plus melphalan vs. melphalan, reported ≥ VGPR, CR and MRD negativity (by NGF) rates of 66.6%, 33.4% and 35% after induction, respectively [69,70]. The other first option as upfront therapy recommended for TE patients is Dara-VTD (daratumumab, bortezomib, thalidomide, dexamethasone) regimen, approved after results from the phase III CASSIOPEIA trial [2] comparing VTD vs. Dara-VTD as induction (4 cycles) and consolidation (2 cycles) after ASCT. Adding daratumumab to VTD increased response rates (at least VGPR after consolidation 83% vs. 78%, *p* = 0.024; at least CR 39% vs. 26%, *p* < 0.0001) and quality of response, being 64% (vs. 44%, *p* < 0.0001) the rate of MRD negativity at level of 10^−5^ in patients receiving Dara-VTD. The last update of the CASSIOPEIA study [71] showed, after a follow-up of 44.5 months, a median PFS not reached in Dara-VTD group vs. 51.5 months in VTD one (HR = 0.58, *p* < 0.0001) with no difference in term of OS, not reached in both arms. Daratumumab has been explored in combination with other triplets as VRD, KRD (carfilzomib, lenalidomide, dexamethasone) and IRD (ixazomib, lenalidomide, dexamethasone). We are waiting for the results of Phase III EMN PERSEUS trial of Dara-VRD vs. VRD as induction and consolidation in 690 TE patients, but randomized phase II GRIFFIN study has already shown that Dara-VRD leads to a 55% reduction in the risk of disease progression and death compared with VRD since, after a median follow-up of 49.6 months, 3-year PFS was 89% in Dara-VRD group vs. 80.7% in VRD one (HR = 0.45, *p* = 0.0324) [72]. In the MASTER phase II study [73], in which TE patients received 4 Dara-KRD cycles as induction, ASCT and up to 8 cycles of Dara-KRD as consolidation on the basis of MRD status, 80% reached MRD < 10^−5^ comparable with 71% obtained after 8 cycles of Dara-KRD in the phase 2 MANHATTAN trial [74]. The randomized phase II ADVANCE study, comparing Dara-KRD vs. KRD, will better clarify the role of daratumumab added to KRD. Quadruplet Dara-IRD has been explored in the phase II IFM 2018-01 study including only standard risk cytogenetics patients and reporting a rate of MRD negativity (10^−5^) of 51.4% after consolidation with 4 cycles of Dara-IRD (preceded by 6 cycles as induction and ASCT) [75]. Comparable results have been obtained adding the other anti-CD38 mAb isatuximab in the quadruplets used as upfront therapy in TE patients. The Phase III GMMG-HD7 trial [76] reported a significantly higher MRD negativity at the end of induction with Isa-VRD vs. VRD (50% vs. 36%, respectively; OR = 1.82, *p* = 0.00017), whereas no data are currently available of the phase III EMN24-IsKia trial comparing Isa-KRD vs. KRD as induction and consolidation after ASCT. However, in contrast with results from GMMG-HD7 study, adding elotuzumab, the anti-SLAMF7 mAb, to VRD, did not improve quality of response and PFS in the GMMG-HD6 trial [77].

In NTE patients continuous therapy with Rd or VMP/VRD regimens represented the standard of care in the previous ESMO guidelines [78]. In the randomized phase IV REAL trial, Bringhen et al. [79] compared PFS of VMP vs. Rd in a real life unselected patient population and no PFS difference was found since median PFS was 29.6 months with VMP and 26.2 months with Rd (HR = 0.82, *p* = 0.41). Currently, daratumumab-containing regimens, together with VRD regimen, have become the main therapeutic options also in NTE patients [68], after results from MAIA [4] and ALCYONE [3] trials. With regards to the former study, after a follow-up of 64.5 months, median PFS was 61.9 months with D-Rd vs. 34.4 months with Rd (HR = 0.55, *p* < 0.0001), with 66.6% of patients alive at 6 years (88.9% of those achieving MRD negativity) after a median follow-up of 73.6 months [80]. In the updated analysis of ALCYONE trial [81], adding daratumumab to VMP continues to prolong PFS (median 37.3 months vs. 19.7 months for D-VMP and VMP, respectively; HR = 0.43, *p* < 0.0001) and OS (median 82.7 months vs. 53.6 months, respectively; HR = 0.63, *p* < 0.0001) after a median follow-up of 78.8 months. The recent phase III IFM 2017-03 trial [82] suggests that the combination daratumumab plus lenalidomide (DR), a dexamethasone sparing-regimen, also improved response rate compared with Rd (ORR = 96% vs. 85%, respectively; *p* = 0.001) in NTE patients with IFM frailty score ≥ 2. An interim analysis at 12-months therapy showed a significantly higher ORR in patients receiving DR vs. Rd (96% vs. 5%, *p* = 0.001), with 64% vs. 43% achieving at least VGPR. Result from IFM 2020-05/BENEFIT trial comparing Isa-Rd vs. Isa-VRD lite in non-frail NTE patients are awaited [83], whereas PETHEMA group will explore iberdomide plus dexamethasone vs. daratumumab, iberdomide, dexamethasone in the GEM-IBERDARAX study. Moreover, phase III Majestec-7 study is comparing teclistamab in combination with daratumumab and lenalidomide (Tec-DR) vs. DRd.

## 4. Autologous Stem Cell Transplantation in Light of New Therapies

The introduction of increasingly more effective regimens has questioned the role of upfront ASCT in TE patients and several trials tried to address the issue. Considering those most recently published, the phase III IFM 2009 study [84] demonstrated that in patients receiving 3 VRD cycles as induction followed by ASCT and 2 VRD cycles as consolidation, median PFS was significantly longer if compared with that of patients treated with 8 cycles of VRD, being 47.3 months vs. 35 months (HR = 0.70, *p* = 0.0001). Phase III US DETERMINATION trial [85], with the same design of IFM 2009 study except for the duration of lenalidomide maintenance that was one year in the IFM trial and until progression in the DETERMINATION one, confirmed the superiority of upfront ASCT in TE patients in term of PFS. The EMN02/HO95 trial [86] in which patients were randomized after a 3–4 VCD induction regimens to receive either 4 cycles of VMP or ASCT, demonstrated the superiority of ASCT over VMP with median PFS of 56.7 vs. 41.9 months, respectively (HR = 0.73, *p* = 0.0001). Using carfilzomib instead of bortezomib in the triplet induction regimens, KRD plus ASCT was compared with KRD without ASCT and with KCD plus ASCT in the phase III UNITO-MM-01/FORTE trial [87]. A significantly higher proportion of patients who received KRD vs. KCD as induction (4 cycles) achieved at least VGPR, so the primary endpoint of study was met (70% vs. 53%, OR 2.14, *p* = 0.0002). Remarkably, after a median follow-up of 51 months among patients who were treated with KRD as induction and consolidation (4 cycles), those who underwent ASCT had a significantly longer PFS compared to those receiving additional 4 cycles without ASCT (KRD12) (4-year PFS = 69% vs. 56%, HR = 0.61, *p* = 0.0084), whereas no significant PFS difference was found between KCD plus ASCT vs. KRD12. The last recently published randomized phase II CARDAMON study [88] showed 2-years’ PFS of 75% in the ASCT group vs. 68% in the KCD group, failing to demonstrate a non-inferiority of KCD compared with ASCT. However, of the above-mentioned studies, only EMN02/HO95 trial reported a significant benefit of ASCT over VMP consolidation in term of OS since after an extended median follow-up, 75-month OS was 69% in the ASCT group vs. 63% in the VMP group (HR = 0.81, *p* = 0.034). It is reasonable to think that the use of induction/consolidation therapy regimens, that are more effective than VCD/VMP used in the EMN study, could make the role of early ASCT not so certain. The development of innovative BCMA-targeted immunotherapies as CAR T cells and bispecific antibodies, recently approved for RRMM patients, paved the way for exploring them in upfront setting. In the ongoing phase III EMagine/CARTITUDE-6 trial [89], that is comparing a cellular therapeutic approach with CAR T cells with conventional ASCT, NDMM patients are randomized to receive either 6 cycles of Dara-VRD followed lymphodepletion and a single cilta-cel infusion or 4 cycles of Dara-VRD as induction followed by ASCT and 2 cycles of Dara-VRD consolidation. In patients aged 60–75 years old, the phase III DSMM XIII study [90] compared continuous lenalidomide plus dexamethasone (Rd) with Rd as induction followed by ASCT with MEL140 and lenalidomide maintenance. The primary endpoint, median PFS, was not significantly different between two arms (38 months for Rd vs. 32 months for ASCT, HR = 1.15, *p* = 0.32) as well as no difference was reported in term of OS. However, among patients enrolled in the ASCT arm, only 66% received a transplant and their median PFS was 40 months, a value much lower than that reported with Dara-Rd in the MAIA trial [4]. In MM older population a longer PFS can be obtained using quadruplet induction (Dara-KRD) and consolidation after ASCT as showed by a post-hoc analysis of MASTER trial [91] comparing the outcomes of patients age ≥ 70 years vs. <70 years. Similar rates of ≥grade 3 adverse events were observed in the two populations and there was no protocol therapy discontinuation due to toxicities. After a median follow-up of 36 months, 3-year PFS was 86.3% and 80.3%, in younger and older patients, respectively (*p* = 0.74), whereas 3-years’ OS was 95.8% and 88.7%. (*p* = 0.53).

## 5. Pros and Cons of Continuous vs. Fixed-Duration Therapy

In both TE and NTE MM patients, long-term therapies aim to prolong the duration of response, allowing to delay occurrence of a relapse that represents a virtually unavoidable event in the course of this haematologic disease. Continuous frontline therapy represents one of the ways to apply this therapeutic approach. In patients without intent for immediate ASCT, US SWOG 0777 trial [92] compared 8 VRd cycles vs. 8 Rd cycles as induction followed by Rd maintenance until progression (median duration of Rd maintenance was 17.1 months). After a median follow-up of 84 months, median PFS was 41 months for VRd and 29 months for Rd (*p* = 0.003), whereas OS was not reached and 69 months, respectively. Remarkably, VRd significantly improved OS in patients younger than 65 years (HR = 0.640, *p* = 0.028), but not in those ≥ 65 years old (HR = 0.769, *p* = 0.168). As described above, longer PFS has been reported with continuous D-Rd evaluated in the MAIA trial [4], showing an improved PFS vs. Rd in all subgroups of patients including those ≥ 75 years in whom median PFS was 54.3 vs. 31.4 months for Rd [80]. Unlike D-Rd regimen, representing a continuous therapy, D-VMP regimen, explored in the ALCYONE study [3], is characterized by an induction with 9 cycles of D-VMP (vs. VPM) followed by daratumumab monotherapy administered every 4 weeks until progression, leading to a median PFS of 37.3 months, definitely lower than that reported with continuous D-Rd. In TE patients, a continuous therapy can be applied with consolidation and maintenance after ASCT. It is undeniable that using as consolidation the same triplet or quadruplet combinations administered as induction, quality and depth of response can be significantly improved as reported by recent trials. In the phase III PETHEMA/GEM2012 trial [69], comparing IV busulfan plus melphalan vs. melphalan as conditioning regimes, the CR rate increases from 33.4% after 6 VRD cycles to 50.2% after 2 VRD consolidation cycles after ASCT with MRD negativity at level of 3 × 10^−6^ went from 28.8% to 45.2%. In the randomized phase II FORTE study [87], after 4 cycles of KRD consolidation after ASCT, at least CR rate and MRD negativity (MCF 10-5) were documented in 54% and 62% of patients, respectively. Switching to quadruplets, in the CASSIOPEIA trial [2] at least CR rate increases from 14.4% after 4 cycles of Dara-VTD as induction to 39% after 2 Dara-VTD consolidation cycles. Moreover, in the GRIFFIN trial, CR rate improved over time in the experimental arm, being 19% after induction and 52% after 2 consolidation cycles of Dara-VRD, whereas MRD negativity at level of 10–5 was documented in 22% and 50% of patients, respectively [72]. However, only two prospective randomized trials evaluated the role of consolidation in the transplant setting and their results were contrasting. After a median follow-up of 74.8 months, the phase III EMN02/HOVON95 trial [93] showed a significantly longer PFS in patients who, after ASCT or VMP intensification, were randomized to receive a consolidation with 2 VRD cycles vs. no consolidation (median 59.3 months vs. 42.9 months, HR = 0.81, *p* = 0.016). On the contrary, the phase III BMT CTN00702 STaMINA trial [94] did not show a PFS benefit in patients who, after an induction therapy predominantly with VRD regimen up to 12 months, were allocated to receive 4 VRD cycles as consolidation plus lenalidomide maintenance vs. double ASCT and lenalidomide maintenance vs. single ASCT plus lenalidomide maintenance. After a median follow-up of 76 months, 5-year PFS was 44.1%, 47.5% and 45% (*p* = 0.685), respectively, with no difference in term of OS [95]. In regard to maintenance post ASCT, continuous lenalidomide represents the standard of care after the phase IFM-2005-02 III [96], GIMEMA RV-MM-PI-209 [97], CALGB 100104 [98] trials and a meta-analysis by McCarthy et al. [99] including these studies showed a significant longer PFS in patients receiving lenalidomide vs. observation/placebo (median PFS 52.8 vs. 23.5 months, HR 0.48). The Myeloma XI trial and a second meta-analysis [100] confirmed these data, reporting HR of 0.47 and 0.72 for PFS and OS, respectively. Comparing results from the IFM 2009 study [84] with those from the US DETERMINATION trial [85]—originally designed as a parallel study to the IFM 2009 but amended for using lenalidomide maintenance until progression instead of for 1 year—patients enrolled in the first trial had a median PFS 20.2 months longer than those receiving lenalidomide for 1 year (67.5 vs. 47.3 months). However, a significant increase in second primary malignancy has been recently reported in ASCT patients receiving lenalidomide maintenance vs. observation in the Myeloma XI trial (cumulative incidence at 7 years: 12.2% vs. 5.8%, *p* = 0.006) [101], so the issue on the best duration of maintenance therapy post ASCT is very relevant. In the TE pathway of Myeloma XI trial, Pawlyn et al. [102] recently reported that the highest benefit of lenalidomide maintenance was observed in the landmark analysis at 2 years from randomization (HR 0.51, *p* < 0.001), at 3 years (HR 0.47, *p* < 0.0001) and 4 years (HR 0.56, *p* = 0.031), but the benefit was no longer significant at subsequent time points. Daratumumab for a maximum of 2 years has been explored as maintenance post ASCT in the phase III CASSIOPEIA trial [103] but a significant benefit in term of PFS was reported only for groups previously receiving induction/consolidation with VTd and not with D-VTd. Results of the combination daratumumab plus lenalidomide (D-R) as maintenance, explored in the GRIFFIN trial, seem to be more promising and phase III trials as PERSEUS and AURIGA will clarify the role of D-R combination post ASCT. In the phase III FORTE trial [87], carfilzomib plus lenalidomide (K-R) significantly improved PFS compared with lenalidomide alone (3-year PFS 75% vs. 65%, respectively) whereas the addition of ixazomib to Rd did not translate in a PFS benefit vs. Rd in the Spanish GEM2014MAIN trial [104]. In contrast, preliminary data from phase III ATLAS trial [105] suggested that a maintenance with a triplet including lenalidomide (KRd) can improve PFS if compared with lenalidomide alone (HR = 0.51, *p* = 0.012). Furthermore, in the maintenance setting, some studies will evaluate novel immunotherapies and the phase III MajesTEC-4 trial will compare the efficacy of teclistamab plus lenalidomide vs. lenalidomide in patients who have completed induction therapy followed by ASCT.; phase III MagnetisMM-7 trial exploring elranatamab vs. lenalidomide after ASCT.; and phase II EMN 26 evaluating 3 different dose levels of iberdomide maintenance post ASCT.

## 6. High Risk Multiple Myeloma and Risk-Adapted Therapies

The extreme biological and clinical heterogeneity of MM makes this disease very difficult to treat in different patient groups. As described above, the presence of t(4;14), t(14;16) and del(17p), taken into account in the R-ISS score [10], to which it can be added 1q gain/amplification (1q21+) and del(1p) identifying patients with HR MM characterized by a median OS ranging from 3 to 5 years [14]. It has to be emphasized that, unlike t(11;14) for which venetoclax, a BLC2 inhibitor, demonstrated significant efficacy in RRMM setting [106], for patients with all other HR cytogenetic abnormalities currently there are no specific available therapies. However, relevant conclusions can be drawn from retrospective analyses of carried out trials and from ongoing risk-stratified studies. In regard to triplet induction regimens, in the phase III DETERMINATION trial [85], median PFS of HR patients who underwent VRD induction and consolidation after ASCT was 55.5 vs. 82.3 months in SR. In the pre-planned cytogenetic subgroup analysis of FORTE trial [107], KRD plus ASCT resulted in higher rate of 4-years’ PFS compared with KCD plus ASCT and KRD12 in all cytogenetic risk groups. Noticeably, in patients receiving KRD plus ASCT 4-years PFS was 82% vs. 67% (HR = 1.89, *p* = 0.11) in patients with 0 or 1 high risk cytogenetic abnormalities (defined as the presence of t(4;14), t(14;16), del(17p) or 1q (gain or amp)), respectively. This suggests that this therapeutic approach could abrogate also adverse effect of 1q gain/amplification. In patients with 2 or more high risk cytogenetics abnormalities (HRCA) 4-year PFS resulted to be 55%, significantly lower than that of patients with 0 HRCA (HR = 2.7, *p* = 0.020). Recently, two retrospective studies compared outcome of HR MM patients receiving VRD or KRD induction followed by ASCT. The first study by MD Anderson Cancer Center [108] included 121 patients with HR cytogenetics defined as t(4;14), t(14;16), del(17p) or 1q (gain or amp), who received a median of 4 VRD or KRD cycles followed by ASCT. After a median follow-up of 34.4 months, 3-year PFS was 53.5% and 64% for KRD and VRD group, respectively (*p* = 0.25), with no difference reported for OS, not reached for either group (*p* = 0.30). The second study evaluated 154 NDMM HR patients treated at Memorial Sloan Kettering Cancer Center [109], who, after induction with VRD or KRD, received early ASCT (77 patients) or no early ASCT (77 patients). In the subgroup of patients undergoing early ASCT, 5-year PFS from ASCT was 24% for VRD and 60% for KRD (HR = 0.49, *p* = 0.04) with 5-year OS of 53% and 87%, respectively (HR = 0.39, *p* = 0.09). Patients who received more than six induction cycles had longer PFS and OS in multivariate analysis. Moreover, a recent retrospective analysis from MD Anderson Cancer Center [110] reported median PFS and OS of 22.9 months and 60.4 months, respectively, in 79 MM patients harboring t(4;14) and receiving triplets as induction (mainly VRD regimen) followed by ASCT and maintenance, confirming the poor outcome of these HR patients in the real-life setting.

In regard to quadruplets, in the CASSIOPEIA study [2] a significant benefit of Dara-VTD over VTD was not observed in HR group neither in term of sCR rate after consolidation nor in term of PFS. In an analysis including patients enrolled in the GRIFFIN and MASTER trials [111], better CR rates were documented in patients with 0 or 1 HRCA (91% and 79%, respectively, for Dara-VRD.; 91% and 89% for Dara-KRD) vs. those with ≥2 HRCA (62% for Dara-VRD and 71% for Dara-KRD). A lower 2-year PFS was also observed, being 97 % vs. 64% in patients with 0 HRCA and ≥2 HRCA, respectively, in GRIFFIN trial and 94% vs. 66%, respectively, in MASTER trial. These results suggest that even quadruplets containing daratumumab cannot significantly improve the outcome of ultra-high risk MM. Double ASCT has shown to prolong PFS and OS in high risk patients as reported by EMN02/HO95 trial [86] in which the highest benefit for double ASCT was observed in patients harboring de(17p). In those patients, double ASCT showed to reduce by 76% risk of progression or death and 5-years’ OS was 80.2% vs. 57.1% with single ASCT. Although long-term PFS and OS of STaMINA trial confirmed no difference in PFS and OS between the three arms, HR patients who underwent double ASCT had a 5-year PFS of 43.7% vs. 37.3% in those receiving ASCT/VRD consolidation vs. 32% in those undergoing ASCT/maintenance (*p* = 0.03) [95], with no OS difference. A recent retrospective, multi-center study by PETHEMA/Spanish Myeloma Group (GEM) [112] compared outcome of HR patients who underwent single or double ASCT in a real-world setting. No difference in terms of PFS and OS was observed but patients with del(17p) receiving single ASCT had median PFS of 41 months while 52% of patients receiving double ASCT were alive and disease free at 48 months. Concerning maintenance post ASCT, in a recent analysis of Myeloma XI trial [113] continuous lenalidomide was associated with marked improvement of PFS and OS in patients with isolated risk markers as del(1p), del (17p) or t(4;14), whereas there was no benefit for patients with gain(1q) and limited benefit for double hit patients for whom new maintenance strategies are needed. In the FORTE trial [107], compared with lenalidomide alone, K-R maintenance was found to reduce the risk of progression or death either in patients with 0 HRCA (3-year PFS: 90% vs. 73%) or in those with 2 or more HRCA (3-year PFS from second randomization: 67% vs. 42%).

Concerning NTE patients, in the MAIA trial [80], treatment with D-Rd significantly improved PFS vs. Rd either in patients with 0 HRCA (*p* < 0.0001) or in those with 1 HRCA (*p* = 0.0003), whereas PFS was similar with D-Rd vs. Rd in patients with ≥2 (*p* = 0.8477). However, in patients with isolated gain/amp (1q21), D-Rd performed better than Rd.

Various recent studies have been designed to improve outcome in HR NDMM patients, thus exploring a risk-adapted strategy based on cytogenetics. Phase II IFM 2018-04 [114] is evaluating Dara-KRD as induction (6 cycles) followed by double ASCT, Dara-KRD consolidation (4 cycles) and D-R maintenance in patients with del(17p), t(4;14) or t(14;16). Preliminary results of 48 patients completing induction, showed 91% of patients achieving at least VGPR, 31% at least CR and 46% MRD negativity. PFS at 18 months was 92% whereas OS was 96%. The phase II GMMG-CONCEPT trial [115] investigated in the arm A (127 TE patients), quadruplet Isa-KRd as induction (6 cyles) and consolidation (4 cycles) after ASCT and, in arm B (26 NTE patients), 8 Isa-KRd induction cycles followed by 4 cycles consolidation. In both arms a maintenance with Isa-KR was planned. MRD negativity (NGF 10^−5^) after consolidation was 67.7% and 54.2% in TE and NTE patients, respectively, so the trial met the primary endpoint. Moreover, 73% of TE and 58% of NTE patients achieved at least CR after consolidation and, in a previous interim 2-year analysis, PFS was 75.5% [116]. The phase II single arm OPTIMUM/MUKnine study [117] enrolled 107 patients with ultra-high risk (UHiR) MM (defined by ≥2 HRCA [t(4;14), t(14;16), del(17p), gain (1q), del(1p) or SKY92 GEP signature). They then received up to 6 cycles with Dara-CVRd followed by ASCT, 18 cycles of consolidation therapy (6 with Dara-VRD and 12 with Dara-VR) followed by maintenance with Dara-R. After a median follow up of 41.2 months, median PFS was not reached; PFS and OS estimated at the end of consolidation therapy, 30 months from the start of induction, were 77% and 83.5%, respectively, with 46.7% of patients obtaining MRD negativity after consolidation. Very encouraging were the data presented at the last ASH Meeting, whereby a single infusion of BCMA/CD19 dual-targeting FasTCAR-T cells (GC012F) as frontline therapy in 16 HR patients [118] was used. Response rates were impressive, being at least VGPR 100%, at least CR 88% and MRD negativity 100% with no grade ≥ 3 cytokine release syndrome (CRS) or neurotoxicity observed. However, wider studies and longer follow-up are needed to establish the role of this new frontier of immunotherapy in upfront setting.

## 7. Minimal Residual Disease (MRD) in the Era of New Drugs and MRD-Driven Therapies

The introduction of three-drug and, more recently, four-drug combinations as induction and consolidation post ASCT allowed for the achievement of deep responses never seen before (Table 2). However, despite these results, most patients continue to relapse suggesting that obtaining even a sCR does not lead to a disappearance of disease, low burden of which can be detected by immunophenotypic and molecular methods. Minimal (or measurable) residual disease (MRD) status has emerged as one of the most potent factors affecting PFS and OS in MM. In the meta-analysis by Munshi et al. [119] including 8098 MM patients, obtaining MRD negativity improved PFS (HR = 0.33, *p* < 0.001) and OS (HR = 0.45, *p* < 0.001), with a significant OS improvement seen in all settings of patients (NDMM and RRMM), regardless of cytogenetics, method of MRD measurement or sensitivity thresholds of it. However, the highest improvements in PFS and OS were observed with MRD negativity at level of 10^−6^ (HR = 0.22, *p* < 0.001; HR = 0.26, *p* < 0.001, respectively). Currently, Next-Generation Flow (NGF) that, using an 8-colour 2 tube panel, is able to reach a sensitivity between 10^−5^ and 10^−6^ and represents one of the most appropriate methodologies to detect bone marrow MRD, as recommended by the International Myeloma Working Group [120]. This method requires more than 5 million cells’ sample and assessment within 24–48 h, but it can be done in a few hours. Using time-dependent analysis, patients with undetectable MRD before maintenance post ASCT (NGF at 3 × 10^−6^ limit) had an 82% reduction in the risk of progression or death (HR = 0.18, *p* < 0.001) and an 88% reduction in the risk of death (HR = 0.12, *p* < 0.00) in the PETHEMA/GEM2012MENOS 65 trial [121]. The other recommended approach to assess MRD is Next Generation Sequencing (NGS), showing a sensitivity up to 10^−6^, requiring less than 1 million cells and feasible in both fresh and stored samples. The value of this method has been confirmed by Perrot et al. [122] who demonstrated, in patients MRD negative before lenalidomide maintenance, a significantly prolonged PFS (HR = 0.22, *p* < 0.001) and OS (HR = 0.24, *p* = 0.001) in the IFM 2009 trial. Additionally, during maintenance, longitudinal MRD testing represents a powerful prognostic factor for outcome as reported by de Tute et al. [123], who found that patients with MRD negativity at 3 months and 9 months after ASCT—or those who converted MRD positivity to negativity by 9 months—had the longest PFS and OS in the Myeloma XI trial. In addition to achieving MRD negativity, sustained MRD is becoming increasingly essential. In a pooled analysis of MAIA and ALCYONE trials enrolling NTE patients, a sustained MRD negativity lasting ≥ 12 months was associated with improved PFS compared with MRD negativity lasting less than 12 months or MRD positivity [124]. In a phase II single-center study from the Memorial Sloan Kettering Cancer Center [125], patients with sustained MRD negativity from start of lenalidomide maintenance had no progression disease after a median follow-up of 19.8 months past the 2-year landmark. Remarkably, loss of MRD negativity was associated with a risk of progression or death (HR infinite, *p* < 0.0001) that was higher than that reported for sustained MRD positivity (HR = 5.88, *p* = 0.015), suggesting that, in the future, MRD resurgence could become a criterion for defining relapse and retreat patients.

MS represents a promising method to detect MRD and it has the not negligible advantage to require peripheral blood instead of bone marrow, allowing simple longitudinal evaluations. Patients with MS negativity at different time points (after induction, prior maintenance and one year of maintenance) had an improved PFS in the GMMG-MM5 trial [50]. Noticeably, CR patients who were MS positive before maintenance had a median PFS of 1.7 vs. 4 years in MS negative CR patients (HR = 2.46, *p* < 0.001). Puig et al. [126] compared the results on MRD assessed with bone marrow NGF and peripheral blood quantitative immunoprecipitation mass spectrometry (QIP-MS) at 2 years from start of maintenance (Rd vs. Rd plus ixazomib) in patients enrolled in the phase III GEM2014MAIN trial. The results of both methods were concordant in 85% of cases, MRD positivity assessed by both methods was associated with a significantly shorted PFS compared with that of MRD negative patients and negative predictive value (NPV) of MS was 87%. In patients enrolled in the same Spanish trial, BloodFlow method integrating immunomagnetic enrichment using MACS MicroBeads before NGF, was found to reach a sensitivity down to 10^−8^ to detect MRD in peripheral blood and it resulted in prognostic for PFS [127]. Imaging techniques such as PET/CT have emerged as key methods to detect the presence of residual disease and combination of bone marrow analysis and PET/CT imaging makes evaluation of response more accurate as shown in the CASSIOPET study trial in which patients who were negative for both MRD by NGF and PET/CT had a better PFS compared with patients who were not double negative [128]. However, PET/CT has limited availability, high cost and criteria for interpretation of results are not standardized.

Several studies either explored or are studying treatments tailored according to MRD response. The single arm, multicenter phase II MASTER trial [73,129] is the first to demonstrate the possibility to discontinue treatment without impact on outcome. After 4 cycles of Dara-KRD as induction followed by ASCT, 123 TE patients received 0, 4 or 8 cycles of Dara-KRD as consolidation based on MRD status, assessed by NGS (<10^−5^) after induction, ASCT and during each 4-cycle block of Dara-KRD consolidation. Patients who achieved two consecutive MRD negative assessments, at the above-mentioned time points, discontinued therapy and entered MRD surveillance (MRD-SURE), whereas patients without 2 consecutive negative MRD assessments after consolidation underwent lenalidomide maintenance. MRD negativity after MRD-directed consolidation was 81% and, after a median follow-up of 34.1 months, 3-year PFS was 91%, 87% and 51% in patients with 0, 1 and ≥2 HRCA, respectively, with 3-years OS of 96%, 91% and 75%. Notably, in patients with 0 or 1 HRCA that discontinued therapy after 2 MRD negative results, 2-year PFS was 91% and 89%, respectively. An even higher risk of progression or death has been found in patients with ≥2 HRCA. Despite reaching MRD negativity, 2-year PFS from treatment cessation was 54%, suggesting that in these patients alternative strategies should be explored as consolidation therapy. Ongoing randomized phase II MASTER-2 trial will evaluate, after 6 cycles of Dara-VRD as induction, ASCT vs. 3 additional cycles of Dara-VRD, in MRD negative patients after induction whereas positive patients will undergo ASCT followed by teclistamab plus daratumumab vs. daratumumab plus lenalidomide. In the phase II/III RADAR (UK-MRA Myeloma XV) study [130], which will enroll 1400 patients, treatment will be deescalated after ASCT in patients with MRD negativity whereas those with persistent MRD positivity will be randomized to different regimens ranging from lenalidomide monotherapy to isatuximab, lenalidomide, bortezomib, dexamethasone (Isa-RBorD). This trial could answer two key questions: how to maintain MRD negativity when it has been achieved and how to obtain MRD negativity if not previously achieved. The ongoing phase III IFM MIDAS trial will probably clarify the role of single or double ASCT since patients with MRD < 10^−5^ after induction therapy with 6 cycles of IsaKRD will be randomized to receive 6 cycles of IsaKRD vs. ASCT plus 2 IsaKRD cycles patients with whereas patients with MRD > 10^−5^ will be allocated to ASCT plus 2 IsaKRD cycles vs. double ASCT. MRD-driven approaches are under evaluation also in the maintenance setting. In the phase III PERSEUS trial, patients allocated in the arm D-R as maintenance (vs. continuous lenalidomide) discontinued daratumumab if achieved a sustained MRD negativity lasting at least 12 months after 24 months of maintenance. In the DRAMMATIC trial, patients are randomized to receive lenalidomide vs. D-R post-ASCT and, after 2 years of therapy, MRD positive patients continue maintenance whereas patients MRD negative are further randomized to discontinue or continue therapy. In the EAA171/OPTIMUM trial, after 10–15 months of lenalidomide maintenance MRD positive patients are randomized to lenalidomide plus ixazomib vs. lenalidomide plus placebo.

## 8. Sequential Therapy in Relapsed/Refractory MM in Light of New Immunotherapeutic Strategies

Natural history of MM is characterized by a continuous succession of remissions and relapses, and the approval of new mAbs in frontline setting are introducing the problem of choosing a correct treatment strategy in RRMM setting, since refractoriness status is one of the principal features to be considered for selecting successive therapies [131]. Data from randomized clinical trials are in support of continuous therapy in RRMM setting, like in NDMM patients, improving survival outcomes, while early progression after fixed duration therapy diminishes quality of life due to several relapses and cumulative toxicity.

For non-lenalidomide-refractory RRMM patients, lenalidomide-based regimens may be used, that could be mAb-based (daratumumab-lenalidomide-dexamethasone, elotuzumab-lenalidomide-dexamethasone) or PIs-based (carfilzomib-lenalidomide-dexamethasone, ixazomib-lenalidomide-dexamethasone, bortezomib-lenalidomide-dexamethasone) [132]. Daratumumab-lenalidomide-dexamethasone (Dara-Rd) triplet demonstrated a significant survival benefit vs. the comparator doublet (Rd) in the recent OS analysis of the phase III POLLUX trial. After a median follow-up of 79.7 months, median OS was 67.6 vs. 51.8 months, respectively, independently of previous lines of therapy (LOT) (1–3 in enrolled patients) even if OS benefit seemed lower beyond the third line of therapy, high cytogenetic risk and age, being the triplet beneficial also in patients with ≥65 years [133].

For lenalidomide-refractory and PI-sensitive RRMM patients, mAbs- or PI-based treatments may be employed. Phase III IKEMA trial randomized 302 RRMM patients with a median of 2 prior LOT to receive isatuximab-carfilzomib-dexamethasone (Isa-Kd) vs. carfilzomib-dexamethasone (Kd). Recent updates confirmed the significant benefit of the triplet compared to the duplet, which a median PFS of 35.7 vs. 19.2 months and 33.5% vs. 15.4% MRD negativity, respectively [134]. Facon et al. have recently published a subgroup analysis of IKEMA trial, confirming the PFS advantage of Isa-Kd in early (24.7 vs. 17.2 months, HR = 0.662) rather than in late relapse (42.7 vs. 21.9 months, HR = 0.542). This advantage was confirmed also for the depth of response [135]. Median PFS was 38.2 vs. 29.2 months in Isa-Ks vs. Kd arms, respectively, in the group of patients with 1 prior line of therapy; whereas it was 29.2 vs. 17 months, respectively, in patients who received >1 prior line of therapy [136]. As for safety, the most common, any-grade, non-hematologic AEs in Isa-Kd were infusion reactions (45.8%), diarrhea (39.5%), hypertension (37.9%) and upper respiratory tract infections (37.3%). Isa-Kd has been approved by regulatory agencies for the treatment of RRMM with ≥1 prior LOT. Daratumumab-carfilzomib-dexamethasone (Dara-Kd), recently approved by FDA and EMA, demonstrated a median PFS of 28.6 months compared to 15.2 months of the comparator arm Kd, in the phase-3 CANDOR trial, whose characteristics could be similar to the IKEMA trial ones. The most common AEs in the Dara-Kd group were thrombocytopenia (25% vs. 16%), hypertension (21% vs. 15%) and pneumonia (18% vs. 9%) [137]. Phase III CASTOR trial [138] recently demonstrated a significant OS advantage of the triplet daratumumab-bortezomib-dexamethasone (Dara-Vd) compared to bortezomib-dexamethasone (Vd) alone, in 498 RRMM patients with a median of 2 prior LOT after a median follow-up of 72.6 months. Median OS was 49.6 vs. 38.5 months, respectively, and the benefit of the triplet was confirmed independently from age, cytogenetic risk and lenalidomide-refractoriness, OS being higher in patients with MRD negativity. Moreover, authors showed that the advantage of the triplet was highest for patients who have received 1 prior LOT (HR = 0.56), dropping for whom have received 2 lines (HR = 0.87) and so on for more LOT (HR > 1 for ≥3 lines of therapy). The most common (≥10%) grade 3/4 AEs with Dara-Vd vs. Vd were thrombocytopenia (46.1% vs. 32.9%), anaemia (16.0% vs. 16.0%), neutropenia (13.6% vs. 4.6%) and pneumonia (10.7% vs. 10.1%) [138]. Daratumumab-pomalidomide-dexamethasone (Dara-Pd) is a recently approved triplet by regulatory agencies for the treatment of RRMM with ≥1 previous line of therapy [139]. Recent OS updates from phase III APOLLO trial demonstrated a significant OS advantage for this triplet compared to the doublet pomalidomide-dexamethasone (Pd) with a median OS of 34.4 vs. 23.7 months, in a population of 304 RRMM patients with a median of 2 prior LOT (range 1–5), whose 79.6% was lenalidomide-refractory. The most common grade 3/4 AEs was neutropenia (68% vs. 51%) whereas pneumonia were reported in 15% vs. 8% of patients and lower respiratory tract infections in 12% vs. 9%, respectively [140]. Pomalidomide-based triplet isatuximab-pomalidomide-dexamethasone (Isa-Pd) was approved for the treatment of RRMM patients with ≥2 previous lines of therapy, on the basis of results of the phase III ICARIA trial, having randomized 307 RRMM patients to receive the triplet vs. Pd. Isa-Pd demonstrated a median OS of 24.6 vs. 17.7 months, compared to the doublet, after a median follow-up of 52.4 months. Additionally, PFS2 and TNT showed continuous benefit with Isa-Pd vs. Pd, without inducing more resistant disease refractory to subsequent treatments. The most common grade 3/4 AEs in the isatuximab group vs. the control group were neutropenia (50% vs. 35%), pneumonia (23% vs. 21%) and thrombocytopenia (13% vs. 12%) [141,142]. Pomalidomide has been also combined to elotuzumab, anti SLAMF7 mAb, in the phase III ELOQUENT-3 trial, demonstrating a significant OS improvement over Pd (median OS 29.8 vs. 17.4 months), beyond the PFS advantage already demonstrated (median PFS 10.3 vs. 4.7 months), and it was maintained across all the subgroups [143,144]. Elo-Pd has been approved for the treatment of RRMM with ≥2 prior LOT. Data from a phase-2 study evaluating efficacy of Elo-Pd in daratumumab-exposed RRMM patients, which were <5% in ELOQUENT-3 trial, showed a PFS of 3.7 months, demonstrating lower efficacy of this triplet in daratumumab-exposed patients. Interestingly, patients who received Elo-Pd immediately following progression on daratumumab obtained significantly longer PFS than patients who got Elo-Pd ≥ 1 line after daratumumab failure; however, they were less heavily pre-treated [145]. Daratumumab-refractoriness is the actual primary challenge for clinicians who treat MM patients [5]. Considering that daratumumab-containing regimens currently represent a standard in TE and NTE patients as above-mentioned, the number of these patients is rapidly increasing. There are limited data on the outcomes of patients relapsing after first-line daratumumab-based therapy and mechanisms of resistance are poorly understood, but re-treatment with anti-CD38 seems to be ineffective [146]. Recent data from the French real life EMMY study reported better but not exciting results when patients were re-treated with anti-CD38 in the early lines of therapy (second and third). Further investigations are needed to answer this challenging question [147]. Therefore, despite the impressive results of mAbs in RRMM, patients continue to relapse and have a dismal outcome [148,149]. Consequently, researchers have combined the specificity of mAbs with a cytotoxic drug, creating a sophisticated delivery system to transport a lethal payload directly to the tumor cells. The selected and most developed target has been BCMA, the B cell maturation antigen expressed at high levels in plasma cells and plasma blasts, but not in other tissues.

### 8.1. Antibody-Drug-Conjugate (ADC)

Belantamab mafodotin (belamaf, GSK2857916) is the first-in-class ADC conjugated to monomethyl auristatin-F (MMF) to demonstrate a significant efficacy in RRMM and it has been approved for the treatment of MM patients who have received ≥4 prior lines of therapy, on the basis of the results from phase II DREAMM-2 study [150,151] whose updated data confirmed 32% of ORR in a heavily pre-treated MM population with a MRD negativity rate of 36% among patients who obtained ≥ VGPR. Median duration of response was 12.5 months and median PFS 2.8 months, reaching 14 months in patients with ≥VGPR, with an estimated median OS of 15.3 months. As for safety, belamaf has introduced a new peculiar toxicity, as keratopathy is the most common grade ≥ 3 reported AE. Ocular toxicity was confirmed reversible since 86% of ocular AEs resolved by the end of follow-up. Treatment discontinuation rate due to ocular AEs was only 3%, without impairment of clinical efficacy because of disease response has been maintained also after several months of stopping treatment [150]. Some real-life experiences are emerging about belamaf outcomes and safety all over the world, which confirmed data from DREAMM-2, without any new safety concerns [152,153,154,155,156]. However, results from phase III DREAMM-3 trial led to FDA request to withdraw belantamab’s US marketing authorization since the study, in which patients were randomized to receive Belamaf vs. Pd, failed its primary endpoint of PFS, being 11.2 vs. 7 months, respectively (HR = 1.03). A lot of trials are evaluating belamaf in combination with other drugs, in order to improve results. Impressive efficacy data derived from the combination of belamaf with pomalidomide in the ALGONQUIN study [157,158] in which 72.2% of patients were triple-refractory. ORR was 86% with a median PFS of 15.6 months, which is really interesting data in a population with more than half triple-refractory patients, whose median PFS does not exceed 6 months in real life studies [148]. In the phase III DREAMM-7 study, belamaf was associated with bortezomib-dexamethasone vs. Dara-Vd in RRMM patients who have received ≥1 previous LOT.; however, its results are yet to be published [159]. Belamaf plus pomalidomide-dexamethasone was compared with triplet pomalidomide-bortezomib-dexamethasone in RRMM patients who have received ≥1 prior line of therapy [160]. Data directly comparing belamaf with other novel therapies in triple-refractory RRMM are lacking, but Weisel et al. recently published a Matching-Adjusted Indirect Treatment Comparison (MAIC) considering CARTITUDE-1 versus DREAMM-2, STORM part 2 and Horizon [161]. After adjustment, patients treated with cilta-cel demonstrated at least a 3.1-fold and at least a 10.3-fold increase in the likelihood of achieving an ORR or ≥CR, respectively, at least a 74% reduction in the risk of disease progression or death, and at least a 47% reduction in the risk of death. About that, the phase III MonumenTal-5 trial has been presented at the last ASH meeting and it will enroll RRMM patients with ≥4 previous lines of therapy to receive belamaf vs. talquetamab until disease progression. Other trials are ongoing, combining belantamab mafodotin with pembrolizumab (DREAMM-4) [162], with novel agents (DREAMM-5) [163], with lenalidomide or bortezomib (DREAMM-6) [164], and with RVd in NDMM setting (DREAMM-9) [165]. Belamaf employment is moving in the frontline setting, where it is administered with lenalidomide-dexamethasone in a phase 1/2 trial [166] or with bortezomib-lenalidomide-dexamethasone [167]. Emerging data from several trials with reduced schedule of belamaf are encouraging to reduce ocular toxicity (DREAMM-9, DREAMM-14, and NCT04808037) [168,169]. Several other ADCs targeting BCMA have been developed (Table 3).

### 8.2. Bispecific Antibodies (BsAbs) and Bispecific T Cell Engagers (BiTEs)

BsAbs are unique constructs binding two targets, one on the tumor cell and the other on an immune cell resulting in a synapsis that enhances immune effector cell activation, proliferation and tumor lysis. Their function also lets the differentiation of naïve T cells to T cells with memory phenotypes (central memory and effector memory T cells), contributing to improved MRD negativity in clinical settings [175]. BiAbs are engineered artificial antibodies, whereas BiTEs are recombinant proteins composed of two linked scFvs (single-chain variable fragment) [176].

Pacanalotamab (AMG420) was the first BiTE to be developed, targeting BCMA and CD3 on T cells; however, despite promising activity, it was overcome early due to its short half-life requiring a continuous intravenous infusion [177]. Pavurutamab (AMG701) derived from the previous BiTE, resolved the half-life problem, being equipped with an extended half-life that gave it the possibility to be administered every 1 to 3 weeks, moving to a subcutaneous administration [178].

Teclistamab (JNJ-64007957) is a BCMA/CD3 BsAbs, it is the furthest along in clinical development and it was approved by FDA and EMA [179]. It demonstrated a 63% ORR (MRD negativity 26.7%) in 165 RRMM patients with at least 3 prior LOT enrolled in the phase I/II MajesTEC-1 trial reporting a median DOR of 18.4 months and a median PFS of 11.3 months. As for safety, it was well tolerated, with 72.1% of CRS (0.6% grade ≥ 3), 70.9% of infections (64.2% grade ≥ 3) and 14% of neurotoxicity (none of which grade ≥ 3) [179]. MajesTEC-1 correlative studies have recently demonstrated that lower baseline T-cell numbers, higher frequency of T-cells expressing PD-1, TIM-3 and CD38 markers, higher frequency of Tregs and CD38+ Tregs, lower proportion of T naïve cells and more NK cells significantly related to an emerging non-responders profile. These data encouraged the association of teclistamab with anti-CD38 mAbs or checkpoint inhibitors [180]. Teclistamab was administered in association with daratumumab-lenalidomide in the phase Ib MajesTEC-2 trial showing a promising ORR with CRS occurring in 81.3% of patients [181].

Elranatamab (PF-06863135), a humanized anti-BCMA/CD3 bispecific IgG2a antibody, demonstrated an ORR of 64% in RRMM patients with 6 median prior LOT whose 91% was triple-refractory, in the phase I MagnetisMM-1 trial. MRD negativity rate was interestingly 100% of evaluable patients and 62% of evaluable patients had sustained MRD negativity at >6 months. As for safety, CRS of grade 1/2 was reported in 67% of patients [182]. Phase-2 MagnetisMM-3 trial cohort A, evaluating elranatamab in RRMM with 5 prior LOTs, showed a 61% ORR with a 90.4% probability of maintaining it at 6 months. New safety concern was 31.7% of grade 3/4 infections, mostly in the upper respiratory tract) [183]. Elranatamab was also used in the phase Ib/II non-randomized umbrella MagnetisMM-4 trial, in association to the gamma-secretase inhibitor nirogacestat in sub-study A and lenalidomide-desametasone in sub-study B, in the setting of RRMM with ≥3 previous LOT. Data are not yet available [184]. Phase III MagnetisMM-5 trial randomized RRMM with ≥3 prior LOT to receive elranatamab vs. elranatamab-daratumumab in part 1, adding the third arm with daratumumab-pomalidomide-desametasone in the part 2. Recent data from part 1 did not report new safety concerns, and described promising response data [185]. Elranatamab was granted an Orphan Drug Designation by the FDA and EMA for the treatment of MM. Other BsAbs in development are detailed in Table 4.

New targets have recently been developed, like G-protein coupled receptor family C group 5 member D (GPRC5D), that are highly expressed on malignant plasmacells and lowly on hair follicles, but not in other tissues [193].

Talquetamab (JNJ-64407564) demonstrated in preclinical studies to induce T cell activation and degranulation, MM cells lysis, IFN-g, TNF-a, IL-2 and IL-10 cytokines activation [194,195]. Talquetamab showed a 73% ORR in 143 RRMM patients with median of 5 prior LOT) enrolled in the phase I/II MonumenTAL-1 trial [196,197], with comparable ORR between triple-refractory or penta-refractory and the ITT population. Median DOR was 9.3 months and median PFS 7.5 months. As for safety, grade ≥ 3 CRS was reported in less than 3% of patients and grade ≥ 3 infections in about 17%. The novel toxicities of this drug were skin-related, nail disorders and dysgeusia occurring in 56%, 52% and 48% of patients, respectively [196,197]. These clinical benefits were confirmed by patient-reported outcomes which endorsed the improvement of quality of life [198]. FDA granted talquetamab a Breakthrough Therapy Designation (BTD) in July 2022 for the treatment of RRMM adult patients who have previously received at least four prior LOT. The ongoing phase III MonumenTAL-3 trial is randomizing RRMM patients with ≥1 prior LOT to receive talquetamab-daratumumab with/without pomalidomide vs. daratumumab-pomalidomide-dexamethasone [199]. Another ongoing phase III trial comparing talquetamab to belantamab mafodotin in RRMM patients with ≥4 previous LOT is MonumenTAL-5, whose data will be available in the near future [200].

Fc Receptor Homolog 5 (FcRH5) is a type I membrane protein that is selectively expressed on B cells and tumor plasmacells, at higher levels than in normal plasmacells. Cevostamab (BFCR4350A) is an IgG-based T-cell-engager BsAb, targeting the most membrane-proximal domain of FcRH5 on MM cells and CD3 on T cells. It demonstrated promising durable responses for ≥6 months and ≥12 months after completion of therapy, in a phase I trial enrolling heavily pretreated patients [201,202].

BsAbs targeting CD38 are exploring in many trials, consequently to the great success of anti-CD38 mAbs, daratumumab and isatuximab (Table 4). Some preclinical studies have shown promising results on CD138/CD3 novel BsAbs [203,204].

Despite reported antigen loss rate seems to be low, resistant RRMM patients who relapsed during therapy with BsAbs/BiTE are described. So, the development of new strategies is ongoing to overcome this. Trispecific antibodies (TriAbs) are novel constructs formed by a single-chain Fv against CD16, which activates NK cells augmenting their cytotoxicity and cytokines production, and two-tumor associated antigens, which increase the specificity to MM cells also increasing the safety [205]. Alternatively, TriAbs could be built to target T cells by two different antigens, like CD3 and CD28, enhancing their activation [206].

### 8.3. CAR-T Cell Therapy

CAR-T cell therapy is a novel approach that has demonstrated promising efficacy in the treatment of RRMM, on the principle of reprogramming the patient’s own T cells to target tumor cells without the physiological need for HLA presentation. Two BCMA CAR-T products have been approved by FDA and EMA for the treatment of triple-class exposed RRMM who have received at least 4 prior LOT, idecabtagene vicleucel (ide-cel, bb2121, ABECMA™) and ciltacabtagene autoleucel (cilta-cel; former LCAR-B38M, CARVIKTY™).

Ide-cel was the first approved by FDA in March 2021, on the results of KARMMA-2 multicohort, phase II, multicenter trial, that demonstrated excellent responses in the majority of 128 heavily pre-treated patients who had received a median of 6 prior anti-myeloma therapies (range, 3–16), with a manageable safety profile [207]. A recent update has reported a median PFS of 12 months in the cohort of patients receiving target dose of 450 × 10^6^ CAR-T cells, ranging from 1.8 months in non-responders to 20.2 months in patients who obtained CR/sCR [208], confirming a strong correlation between the depth of response and outcomes. Ide-cel demonstrated high quality response (85% MRD negativity) in patients with early relapse after frontline therapy with median PFS of 11.4 [209] and in patients who have had an inadequate response to frontline autologous stem cell transplant (HSCT) (74.2% of CR/sCR) [210]. Ide-cel demonstrated to have a deep clearance of tumor burden in this challenging setting, with patients who obtained ≥ CR and sustained it for more than 2 years. But the rates of functional “cure” remain modest at best with current treatment strategies, as evidenced by almost half of the patients on the KARMMA study who progressed within the first year of treatment even at the target dose of 450 × 10^6^ CAR-T cells, remarking the critical role of high-risk features on prognosis.

Cilta-cel is a differentiated CAR-T therapy with two BCMA-targeting single-domain antibodies to confer it avidity [211]. It has been recently approved on the basis of the results of phase Ib-II CARTITUDE-1 trial which demonstrated an ORR of 97.9%, of which 82.5% was sCR, in a MM population who had received more than 3 prior lines of therapy or were double refractory [212]. Munshi et al. recently showed some high-risk patients’ characteristics, like high-risk cytogenetic, ISS stage, number of prior lines of therapy and penta-drug refractoriness, as predictive of the achievement of sustained MRD negativity at 6 and 12 months, which correlate with better outcomes. The presence of baseline extramedullary plasmacytomas was less common in patients with sustained MRD negativity [213]. These preliminary descriptive data paved the way to more specific studies on the role of CAR-T cell therapy in high risk RRMM. Cilta-cel showed 75% CR/sCR in the cohort A of CARTITUDE-2 trial [214], including patients with progressive disease after 1–3 lines of therapy. Responses were early (median time to first response of 1 month) with a manageable safety profile in patients treated in an outpatient setting [214]. In the cohort B of the study, enrolling 19 patients with early relapse after initial therapy, ORR was 100% with 90% of CR/sCR. After a median follow-up of 18 months, cilta-cel demonstrated durability and deepening of response (93% of MRD negativity), maintaining the 12-months PFS rate of 90% [215]. As for safety, CRS was always reversible, only 1 grade 4 supporting further exploration of the agent in earlier lines of therapy. For both the CAR-T products, cytopenias were frequent and not dose related, with a median time to recovery from grade 3 or higher neutropenia and thrombocytopenia of 2 and 3 months, consisting of a challenging new issue.

Phase 1 and 1/2 trials are exploring other BCMA-directed CAR-T cell products, all but one being autologous [216]. Despite the impressive results of a “one-shot” therapy, never seen so far, CAR-T cell therapy failed to heal MM patients. A scientific solution has been searched in the host and/or in the manufacturing of the CAR-T product. Improving patient selection seems to be the most immediate need, to overcome both toxicity and resistance [217]. Selecting patients on the basis of hypothetical specific features predicting a deep and durable response could spare this therapy for patients—who had contrarily non-responders’ features—and customize therapeutic approach for patients with a responder profile. Taking into account this last item, trials are employing CAR-T in early relapses (KarMMa-3, NCT03651128; CARTITUDE-4, NCT04181827) or frontline (BMTCTN1902, NCT05032820; CARTITUDE-5, NCT04923893; CARTITUDE-6, NCT04181827), suggesting that CAR-T cell therapy in earlier lines of therapy could be safe and may yield high and deep responses in different unmet-need populations. Moreover, between 10% and 25% of the patients who underwent apheresis in the different trials did not reach infusion due to complications during the manufacturing time or disease progression. It is not an off-the-shelf therapy, making the aggressive relapses difficult to manage. Therefore, platforms for rapid CAR-T cell manufacturing or universal off-the-shelf allogeneic CAR have been developed [216]. Potential mechanisms to overcome resistance to CAR-T are listed in Table 5 and several real-life experiences on CAR-T therapy presented at the last ASH Meeting are listed in Table 6.

### 8.4. Sequencing Novel Immunotherapies in Advanced MM

Upcoming anti-BCMA immunotherapies are paving the way to the challenge how to place them into the therapeutic landscape in MM. The observation of disease relapses during anti-BCMA therapies and the lack of effective therapeutic strategies afterwards have fostered some research on the efficacy of re-treating patients with the same class of drugs. Currently, a definitive rule of optimal sequence cannot be given, but when CART are used after BsAbs or ADC, the length of exposure to anti-BCMA and the free interval from BsAb/ADC to CART could be considered the most important predictive factors of response to re-treatment [230]. ADC followed by BsAbs-CART is the most frequent utilized anti-BCMA sequence because ADC were available all over the world before than BsAbs/CART. There are a few experiences with the opposite sequence. Teclistamab was administered after ADC or CART in cohort C of MajesTEC-1 trial, obtaining a 52.5% ORR without difference between ADC and CAR-T and with similar safety profile than in BCMA-naive patients [231]. MagnetisMM-3 trial are evaluating in the cohort B elranatamab efficacy and safety in RRMM patients already exposed to ADC or CART, but data are awaited [232]. Moreover, novel drugs recently gave the chance to change the target, using GPRC5D or others, adding another piece to the complex therapeutic sequence. The results of these sequences are controversial and required more data. Interestingly, a recent real-life experience showed that patients previously exposed to anti-BCMA products, high-risk cytogenetics, poor performance status and younger age have had worse outcomes [233]. Therefore, finding the best anti-BCMA sequencing strategy could let clinicians dramatically improve RRMM outcomes and benefits; however, a perfect strategy currently does not exist and further trials are encouraged to find it.

## 9. Precision Medicine and Next Generation Therapies

Venetoclax (Ven) is a potent and selective oral BCL-2 inhibitor with demonstrated anti-myeloma activity in pts with t(11;14), thus making it the first example of precision medicine in MM.

The phase III BELLINI trial, randomizing RRMM patients with 1–3 prior lines of therapy between venetoclax-bortezomib-dexamethasone vs. bortezomib-dexamethasone showed better outcomes for the venetoclax-bortezomib-dexamethasone arm, although increased mortality was observed in the venetoclax group, reflecting a higher incidence of death related to infection [106]. A phase 1/2 trial is evaluating venetoclax in association to dara with/without bortezomib in t(11;14) RRMM patients, part 3 of this study has been recently updated demonstrating an ORR of 95%, 100% and 62% for the Ven400Dd, Ven800Dd, and DVd arms, whereas the ORR for the combined Ven arms was 98%. VenDd demonstrated deep responses that appear to be durable; data are not mature and follow-up is ongoing [234].

Iberdomide (CC-220) is an orally available CELMoDs (cereblon E3 ligase modulator) agent that binds to the cereblon E3 ubiquitin ligase complex leading to greater degradation of Ikaros and Ailos than lenalidomide and pomalidomide. It has been investigated in the phase I/II CC-220-MM-001, with 31.9% ORR and a manageable safety profile, and phase III EXCALIBER-RRMM studies with different treatment combinations in RRMM patients. Iberdomide-dexamethasone combination recently showed encouraging efficacy and safety in patients with triple-class-exposed (ORR 36.8%) or triple-refractory RRMM and prior anti-BCMA therapy in specific cohorts of CC-220-MM-001 trial [235].

Mezigdomide (CC-92480) is another potent CELMoD, that has a significantly higher degradation efficiency compared to either lenalidomide or pomalidomide, having shown a 55% ORR in a heavily pre-treated RRMM population enrolled in the phase I CC-92480 trial [236]. Efficacy data in triple-class refractory RRMM, including patients with prior BCMA-targeted therapies are promising, showing 40% and 50% ORR, respectively [237]. Selinexor is a first-in-class, selective exportin-1 inhibitor, that is approved in the EU and USA for the treatment of adult patients with MM who have received at least one prior therapy. Approval of the selinexor-bortezomib-dexamethasone (XVd) regimen was based on the phase III BOSTON trial, in which 195 patients received XVd vs. 207 twice-weekly Vd, with a median PFS of 13.93 vs. 9.46 months, respectively [238]. The phase I/II STOMP trial, which showed an ORR of 78% in RRMM patients treated with Xd-carfilzomib, recently confirmed this advantage also in triple-refractory patients (ORR 67%) [239].

Modakafusp alfa (TAK-573) is an example of an immunocytokine designed as a delivery system of interferon alpha (IFNα) 2b to CD38+ cells. It is a recombinant humanized IgG4 anti-CD38 mAb that is fused to an attenuated IFNα protein and binds to a unique epitope of CD38. It demonstrated a 43% ORR in a phase-1 trial that enrolled RRMM patients with ≥3 prior lines of therapy, at the dose of 1.5 mg/kg Q4W. These promising results were confirmed in triple-refractory and penta-exposed patients with a 41% ORR, higher in non-BCMA-exposed than in BCMA-exposed patients (ORR 60% vs. 27%). In BCMA-refractory patients ORR was 25% [240].

## 10. Discussion and Expert Opinion

Upfront therapies, particularly those including mAb as daratumumab, opened a new horizon in the management of NDMM since they led to results never obtained before in both NTE and TE patients. However, although an improved outcome has been reported also in high-risk cytogenetic patients, the new regimens do not seem to overcome the poor impact of ultra-high risk features, defined by the presence of at least two high-risk cytogenetic abnormalities, as reported by recent trials [107,111]. Preliminary results from ongoing studies exploring risk-adapted therapies, mainly based on cytogenetics, are promising but, beyond traditional FISH, it must be emphasized that newer technologies such as whole-exome and whole-genome sequencing as well as GEP can improve prognostication in MM.; Moreover, in the near future they will probably be able to predict responses to treatments, tailoring therapy to risk and creating a personalized medicine approach. Unfortunately, these methods are complex, expensive and currently difficult to apply in routine clinical practice. Similarly, MRD testing by NGF or NGS, proved to be a more powerful factor in affecting PFS and OS than cytogenetics or ISS stage in several studies [121,122], but it suffers from several issues limiting its incorporation in daily practice. Clinical studies did not definitively clarify neither what the ideal threshold of sensitivity is, the optimal timing for MRD assessment, nor if achievement of MRD negativity has the same value in SR, HR or ultra-high-risk patients, given that the MASTER trial [130] showed different outcomes in these groups of patients despite MRD negativity. Moreover, the discordance between the achievement of CR according to IMWG criteria [120] and MRD negativity [241] should be better explained. Another unresolved issue is in regard to the minimal duration of sustained MRD negativity required to be a strong prognostic factor since, as reported in patients enrolled in the MAIA and ALCYONE trials [124], PFS was not the same when sustained MRD lasted less or more than 12 months. However, based on the significant MRD negativity rates obtained with continuous therapies or through the different phases of treatment, sequential MRD evaluation could better assess the efficacy of each treatment phases, allowing therapy escalation or de-escalation with the aim to minimize toxicity and costs. Ongoing phase III trials [88,105,117,129,130] will enable us to understand if MRD-driven approach will be the right tool to personalize therapy. Early results on the comparison between bone marrow data and peripheral blood MRD assessment by mass spectrometry (MS) seem to be promising—and this and other similar techniques may represent a turning point in the clinical practice.

Moving all available agents in the upfront regimens, the number of double- or triple-refractory in early relapse will increase progressively [5] and the treatment of RRMM patients who become refractory to lenalidomide and daratumumab could become a more difficult challenge. Triplets approved for early relapses include anti-CD38 mAbs as daratumumab or isatuximab but data on the efficacy of a retreatment with anti-CD38 mAb are disappointing [147]. Until novel immunotherapies are available in the early lines of therapy, these patients are orphaned of effective regimens. Triple-refractory patients could benefit from new immunotherapies that, currently, have been approved only for patients who have received at least four prior lines of therapy. Nevertheless, in the near future, most patients will be triple- or penta-refractory even before fourth line, making the treatment choice even more challenging. In this scenario, therapy sequencing from first to advanced lines of therapy is becoming a crucial issue to optimize regarding personalized therapy of MM. Several ongoing trials are trying to address this critical issue.

## 11. Conclusions

The landscape of MM treatment is changing quickly and it will continue to do so, as the progress in MM biology will allow us to identify more and more powerful prognostic and predictive factors. The discovery of new agents with peculiar mechanisms of action will also make feasible risk- or response-adapted therapies, minimizing long-term toxicities and improving the quality of life of patients.

## Figures and Tables

**Table 1 cancers-15-02203-t001:** Other risk stratification models.

Risk Score/Parameter	Description	References
**Genomic approaches**		
**IAC-50 Model**	Including both clinic and genomic features (a 46-gene expression signature, beta2-microglobulin, ISS stage, and first-line treatment scheme). It was validated in an external cohort and outperformed UAMS70 in the prediction of OS, particularly in the first 2 years after diagnosis.	[38,39]
**EMC92-ISS**	Four-group model, strong predictor for OS combining ISS and EMC92 genomic signature	[40]
**SKY-RISS**	Combining R-ISS and SKY92 signature and acted as an immunomodulatory agent predictor	[41]
**SKY92 + FISH**	Combining SKY92 and FISH identifying the “highest-risk” MM in a real-life study	[42]
**PCL-like transcriptome + R2-ISS**	Identifying an exceptionally high risk NDMM population, with a median OS of only 7 months	[43,44]
**Clinical-Mixed approaches**		
**Cytogenetic PI**	Improving R-ISS with cytogenetic features as del(17p); t(4;14); del(1p32); 1q21 gain	[45]
**Durie/Salmon plus**	Incorporating EMD and Durie/Salmon staging system	[46]
**CPC + R-ISS**	quantifying cPCs by MFC can potentially enhance the R-ISS classification of a subset of NDMM patients with stage I and II disease by identifying those patients with a worse than expected survival outcome	[47]
**Nomogram Prognostic Model**	Using CTCs as an independent predictor of PFS and OS in NDMM patients. Nomograms predicting PFS and OS were developed according to CPC, lactate dehydrogenase (LDH) and creatinine	[48]

**Table 2 cancers-15-02203-t002:** MRD rates in three-drug and four-drug clinical trials.

Trial	Phase	Regimen	≥CR (%)	MRD Negativity (%)	MRD Method/Sensitivity	Timing of Response Assessment
PETHEMA/ GEM2012	III	VRD (6) → ASCT → VRD (2)	50.2	45.2	NGF/3 × 10^6^	Post-consolidation
FORTE	II	KRD (4) → ASCT → KRD (4)	54	62	MCF/10^−5^	Post-consolidation
CASSIOPEIA	III	Dara-VTD (4) → ASCT → Dara-VTD (2)	39	64	NGF/10^−5^	Post-consolidation
GRIFFIN	II	Dara-VRD (4) → ASCT → Dara-VRD (2)	52	50	NGS/10^−5^	Post-consolidation
MASTER	II	Dara-KRD (4) → ASCT → Dara-KRD (4) → Dara-KRD (4)	86	81	NGS/10^−5^	Post-consolidation (MRD-driven)
IFM 2018-04 (HR)	II	Dara-KRD (6) → ASCT → Dara-KRD (4)	31	61	NGS/10^−5^	Post-induction
IFM 2018-01	II	Dara-IRD (6) → ASCT → Dara-IRD (4)	32.6	39.5	NGS/10^−6^	Post-consolidation
GMMG-HD7	II	Isa-VRD (3) → ASCT	24.2	50.1	NGF/10^−5^	Post-induction
GMMG-CONCEPT (HR)	II	Isa-KRD (6) → ASCT → Isa-KRD (4)	72.7	67.7	NGF/10^−5^	Post-consolidation

**Table 3 cancers-15-02203-t003:** Other ADCs in development.

Name	Description/Target	ORR (%)	References
AMG 224	humanized IgG1 anti-BCMA mertansine-conjugated	27%	[170]
MEDI2228	humanized pyrrolobenzodiazepine (PBD)-conjugated	65.9%	[171]
Indatuximab ravtansine	anti-CD138 chimerized MAb (nBT062) linked to the maytansinoid DM4	-	[172]
Lorvotuzumab mertansine	CD56-binding antibody with a maytansinoid cell-killing agent, DM1, attached using a disulfide linker, SPP	-	[173]
STRO-001	fully human, aglycosylated anti-CD74 ADC incorporating a non-cleavable linker-maytansinoid warhead with a drug-antibody ratio of 2	-	[174]

**Table 4 cancers-15-02203-t004:** Other BsAbs in development.

Name	Description/Target	ORR (%)	References
**Linvoseltamab (REGN5458)**	fully humanized BCMA/CD3, generated by Regeneron’s proprietary human antibody mouse technology (VelocImmune) and full length BiAb platform (VelociBiTM)	75	[186,187]
**TNB-383B (ABBV-383)**	next-generation BCMA/CD3 fully human bispecific monoclonal IgG4 antibody	>50	[188]
**Alnuctamab (CC-93269)**	BsAb with 2 asymmetric arms carrying humanized IgG1 T-cell engagers that bind bivalently to BCMA and monovalently to CD3 in a 2 + 1 format	39	[189]
**RG2634 (RO7425781)**	GPRC5D/CD3-directed BsAb, characterized by silent Fc region that reduces toxicity and increases its half-life	68	[190]
**ISB 1342 (GBR 1342)**	first CD38/CD3 BsAb engineered (using the Glenmark Bispecific Engagement by Antibodies based on the T cell receptor [BEAT] platform)	-	[191]
**AMG 424**	humanized T cell-recruiting CD3/CD38 BsAb	-	[192]

**Table 5 cancers-15-02203-t005:** Potential mechanisms to overcome resistance to CART.

Technique	Description	References
**Improving CART manufacturing**	Cellular platforms such as natural killer CART or γδ T cells try to enhance CART expansion and persistence	[218,219]
**Creating humanized or fully human constructs**	A way to overcome the negative impact of immunogenicity in CART outcomes	[220]
**CART enrichment**	CART enrichment with a T cell subset having a memory-like phenotype and a superior proliferative capacity upon adoptive transfer	[221]
**Developing next-generation CART**	CART directed towards other targets could be the solution to overcome the antigen loss (FASTCAR platform are exploring with a dual BCMA/CD19-targeted CART)	[219]
**Identifying alternative targets**	GPRC5D, SLAMF7, CD138, CD38, light chains and CD19	
**Fine-tuning of CAR density in the T cell**	Molecular refinements to the CAR spacer can impact multiple biological processes and antitumor activity	[222]
**Co-infusion of CAR and chimeric costimulatory receptors (CCR)**	Combination of a CAR and a CCR to improve the clinical outcomes of CART by enhancing cytotoxic efficacy and persistence	[223]
**Using HLA-independent T-cell receptors instead of CAR**	HLA-independent T cell receptors (hit receptors) consistently afford high antigen sensitivity and mediate tumor recognition beyond what CD28-based cars	[224]
**Armoring CART**	Combination of CART with gamma-secretase inhibitors to increase BCMA density in the tumor cell	[225]

**Table 6 cancers-15-02203-t006:** Real life studies on CART.

Trial	Results	References
Multicenter experience (USA, University of Texas)	Ide-cel was associated with a trend towards worse efficacy outcomes for patients who received it less than 6 months after a previous anti-BCMA therapy. In CARTITUDE-2 depth of response and DoR appeared lower in patients already treated with anti-BCMA than in naïve patients (≥VGPR in 55%, 46.7% and 42.9% in full cohort, ADC exposed and BsAb exposed, respectively)	[226]
Multicenter experience (USA, University of Standford)	Feasibility of Ida-cel in 28 patients with renal failure, with the same efficacy data than patients with normal renal function, but significantly higher rate of grade ≥ 3 cytopenia, within 90 days, and longer hospitalization stay	[227]
Multicenter experience (USA, University of California)	Feasibility of Ide-cel in in 77 patients with ≥70 years, with similar outcomes than younger patients and without a significant increase of toxicity	[228]
Multicenter experience (Europe, France)	Efficacy and safety data od Ide-cel was comparable to these of the original study	[229]

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
