# Peer review of "Current Main Topics in Multiple Myeloma"

_cancers, 2023, doi:10.3390/cancers15082203_

Round 1
Reviewer 1 Report
Morè and colleagues report an extensive review of the current main topics concerning multiple myeloma, accurately describing the therapeutic novelties currently in various stages of development and discussing the most recent evidence presented in the literature (even ASH 2022 and over). The paper is thorough and well-described.
However, I suggest to the authors to apply an adequate revision of the English language and style, with the use of shorter and less elaborate sentences in some paragraphs and correcting various grammatical imperfections in such a way as to make reading more fluent and correct.
Author Response
Dear reviewer,
Thanks a lot for appreciating our paper, we provide to review English language and grammar.
Reviewer 2 Report
The review titled “Current Main Topics in Multiple Myeloma” by Sonia Morè et al. discusses various topics on risk stratification, treatment, and management of Multiple Myeloma. Quite a few areas are covered relating to current developments of multiple myeloma, and it is very well written. It is believed that this article can be published as it is.
Author Response
Dear reviewer,
thanks a lot for your good opinion on our paper.
Reviewer 3 Report
Cancers-2251205
Current Main Topics in Multiple Myeloma
The review article “Current Main Topics in Multiple Myeloma (cancers-2251205)” discussed 10 hot topics in the field of myeloma treatment. Contents in all the topics were described in detail, but the volume of manuscript was too much for readers to understand. Therefore, I considered the volume of manuscript could be reduced by focusing several topics. Considering the title, future directions, such as sessions 9 to 11, can be cut, or the volume of those can should be reduced at least, and add the contents about BCMA targeting CART and BiTE into “8. Sequential Therapy in Relapsed/Refractory MM”. Thus, I recommend that the total manuscript should be reconstituted.
Author Response
Dear reviewer,
thanks for your comments. We provide to reduce the volume of manuscript as requested, but we would not minimize the great complexity of the topic. So, we have reduced the contents of chapters 9 to 10 and added them into chapter 8. However, we believe that chapter 11 should be mantained because it deals with a different and important issue.
Round 2
Reviewer 3 Report
I considered that this revised version was suitable for publish for "Cancers".